# Digital Health Interventions for Promoting Healthy Aging: A Systematic Review of Adoption Patterns, Efficacy, and User Experience

Majed M. Alruwaili [1],*, Mostafa Shaban [2],* and Osama Mohamed Elsayed Ramadan [3]

1    Nursing Administration Department, College of Nursing, Jouf University, Sakaka 72388, Saudi Arabia
2    Community Health Nursing Department, College of Nursing, Jouf University, Sakaka 72388, Saudi Arabia
3    Maternity and Pediatric Health Nursing Department, College of Nursing, Jouf University, Sakaka 72388, Saudi Arabia; omramadan@ju.edu.sa
*    Correspondence: majed@ju.edu.sa (M.M.A.); mskandil@ju.edu.sa (M.S.)

**Abstract:** Background: Global population aging poses challenges for healthcare. Digital health technologies may benefit older adults through enhanced access, monitoring, and self-care. This systematic review evaluates the intersection of digital health interventions and healthy aging, focusing on adoption, efficacy, and user experience. Methods: PubMed, Embase, and Cochrane Library were systematically searched for studies on digital health technologies for adults aged 50+ years. Randomized controlled trials, observational studies, surveys, and qualitative studies were included. Outcomes were adoption rates, efficacy, and qualitative feedback. Study quality was assessed using standardized tools. Results: 15 studies were included. Adoption increased during COVID-19, but divisions persist. Barriers like technology challenges and distrust require addressing. Web-based programs and telerehabilitation demonstrated benefits for behaviors and balance. Users had positive attitudes but emphasized patient-centric, ethical design. Most efficacy data were preliminary; more rigorous trials are needed. Discussion: Digital health interventions show promise for supporting healthy aging, but thoughtful implementation strategies tailored to user needs and capacities are essential to realizing benefits equitably. More efficacy research and studies on real-world integration and ethics are warranted. Conclusions: Digital health has significant potential for promoting healthy aging through enhanced access, monitoring, and self-care. However, evidence-based, patient-centered solutions are imperative to maximize adoption, efficacy, and positive user experience for diverse older adult populations.

**Keywords:** digital health; healthy aging; telehealth; telerehabilitation; systematic review

## 1. Introduction

The global population is undergoing a major demographic transition with increasing numbers and proportions of older adults. According to the United Nations, the number of persons aged 60 years or over is expected to more than double by 2050, rising from 1 billion in 2020 to over 2.1 billion [1]. This global aging phenomenon reflects declining fertility rates combined with rising life expectancies worldwide. While population aging represents a public health success, it also poses significant challenges for healthcare systems [2]. Advancing age is the key risk factor for most chronic diseases, geriatric syndromes, and loss of functional independence [3]. As people live longer, their risk of developing age-related conditions like heart disease, cancer, respiratory illness, diabetes, arthritis, dementia, osteoporosis, sarcopenia, and depression increases exponentially [4]. The concept of "healthy aging" has, therefore, gained prominence in recent years, emphasizing the need for evidence-based strategies to maximize health, well-being, and quality of life across the older adult lifespan [5].

Digital health encompasses a wide range of technologies and solutions, including wearable devices, telehealth platforms, mobile health apps, and personalized medicine enabled by big data analytics and AI [6–8]. When designed with senior-friendly principles, these technologies have the potential to enhance multiple aspects of healthcare for older adults [9]. Benefits may include increased access to care, more preventive health monitoring, reduced hospital visits and costs, improved medication adherence, better chronic disease management, and greater patient engagement and empowerment [10,11].

Digital health broadly encompasses solutions like telemedicine, remote patient monitoring using devices and sensors, mobile health applications, and virtual/augmented reality platforms supported by artificial intelligence [12,13]. These innovations open new avenues to overcome barriers hindering prevention, management and rehabilitation services for aging populations [14]. For example, telemedicine can expand specialist access in underserved communities [15]. Mobile apps support chronic disease self-care through reminders and education anywhere, anytime [16]. Continuous remote monitoring allows for tracking health factors at home. Virtual reality may aid rehabilitation by motivating exercises and simulating activities of daily living [17,18].

If designed accessibly and widely adopted, digital health technologies could help address global aging challenges [19]. Benefits may include facilitating aging in place, reducing hospital and institutional resource use, and enabling early detection of health status changes through enhanced infrastructure for remote care, monitoring and therapy [20]. While the body of evidence is growing, limitations persist regarding adoption patterns, efficacy outcomes, and user experiences of digital tools targeting older populations [21]. Individual studies often explore single technologies or conditions rather than synthesizing the broad landscape. Rigor also varies, with many efficacy trials limited by small sample sizes and lack of controls [22]. Research has predominantly focused on perspectives from high-income settings despite barriers faced in lower-resource regions. Qualitative components exploring end-user needs are underrepresented compared to clinical measures. Rapidly evolving solutions require up-to-date evaluations as well [23].

As a result, knowledge remains incomplete around factors influencing adoption among seniors, effective applications across domains, varying user viewpoints considering individual differences, ensuring equitable access, and overcoming challenges linked to age, income, and resources that inhibit use [24]. Large, comprehensive assessments are still needed to establish real-world implementation feasibilities and long-term impacts. In addition, comprehensively assessing available evidence establishes which application areas, user groups, and health conditions have and have not yielded clearly beneficial impacts from existing technologies [25]. The analysis thereby substantiates and prioritizes future research targets around under-explored uses, comparing populations experiencing disparities and unmet needs [26,27]. Greater clarity regarding factors impacting take-up can also empower seniors and caregivers in technology-enabled self-care and empowerment [28].

Also, Results summarize quantitative outcomes across various digital health interventions and domains to elucidate clinically effective options deserving wider implementation consideration and guidance. Qualitative insights further suggest optimization strategies for technology integration respecting staff needs and workflows [29,30]. This equips frontline teams with an evidence-based understanding of the viable role digital tools can play in complementing and extending traditional services [31].

Also, factors influencing adoption, barriers limiting uptake, and target users' expressed needs and preferences when engaging with aging-focused digital solutions. This real-world evidence can guide patient-centered innovation approaches that optimize existing tools and inform new product designs aligned with end-user values, contexts, and desired impacts. Ensuring technologies deliver benefits respectfully meeting psychosocial needs as well as clinical efficacy enhances prospects for market success and sustainability.

This systematic review aims to understand the impact of digital health interventions on adoption rates, efficacy, and user experience among older adults in promoting healthy aging. Specifically, the research question is: What is the impact of digital health interventions on

adoption rates, efficacy, and user experience among older adults (aged 50+) in promoting healthy aging?

## 2. Materials and Methods

### 2.1. Search Strategy and Selection Criteria

The systematic review conducted in this study followed the requirements outlined in the Preferred Reporting Items for Systematic Reviews and Meta-Analysis (PRISMA) guidelines [32], which were meticulously followed to ensure a thorough and transparent methodology. The research protocol for this investigation was developed according to the guidelines specified in the Preferred Reporting Items for Systematic Reviews and Meta-Analysis Protocols (PRISMA-P) statement [33] and was appropriately registered with PROSPERO (CRD42023472296) (Supplementary Materials).

A comprehensive search approach was systematically implemented across reputable databases, such as Embase.com, Medline ALL (Ovid), Web of Science Core Collection, Cochrane Central Register of Controlled Trials (Wiley), and Google Scholar, among others. The latest search was conducted on 25 July 2023. The search method utilized a modified amalgamation of relevant medical topic headings (MeSH) and keywords pertaining to digital health interventions, healthy aging, technology-assisted aging, and associated terminology. Furthermore, a meticulous manual examination of the references cited in the selected publications was conducted in order to guarantee a complete scope.

### 2.2. Eligibility Screening

Following the removal of duplicate entries, a rigorous eligibility screening was carried out by two independent reviewers, namely M.M.A and O.M.E.R. The screening process initially consisted of the evaluation of titles and abstracts, which was subsequently followed by a thorough study of the complete text. The inclusion criteria consisted of research that particularly examined digital health interventions aimed at promoting healthy aging. These interventions comprised many methods, including but not limited to mobile applications, wearable devices, telemedicine, and health monitoring systems. The exclusion criteria encompassed research that examined therapies deemed irrelevant, opinion articles, case reports, and studies conducted in languages other than English. Any inconsistencies that arose during the screening process were thoroughly addressed by consensus reached via discussion among the reviewers.

### 2.3. Data Extraction

The main aim of this systematic review was to thoroughly examine the patterns of adoption, effectiveness, and user perceptions related to digital health interventions in the context of promoting healthy aging. The process of data extraction was carried out by two reviewers in a manner that ensured independence. The focus of this extraction was on crucial aspects of the studies, including key characteristics, intervention specifics, outcome measures, rates of adoption, metrics of efficacy, and comments from users. In instances where it was deemed required, the authors of the chosen papers were proactively reached out to in order to get any further or absent data that were deemed crucial for the purposes of this evaluation. Furthermore, a comprehensive evaluation was conducted to identify and address any potential overlap or duplication in participant cohorts across the studies. This process involved consulting with the authors of each relevant study to ensure accurate resolution.

A total of 4601 documents were identified during the initial search of the databases. Following the removal of duplicates, a total of 488 papers underwent screening based on their title and abstracts, resulting in the exclusion of 90 papers. Out of the remaining 398 papers, Reports were evaluated to determine their eligibility. Out of a total of 176, only 15 were chosen for the complete text evaluation [19,34–47]. Figure 1 provides an explanation of the PRISMA flow diagram.

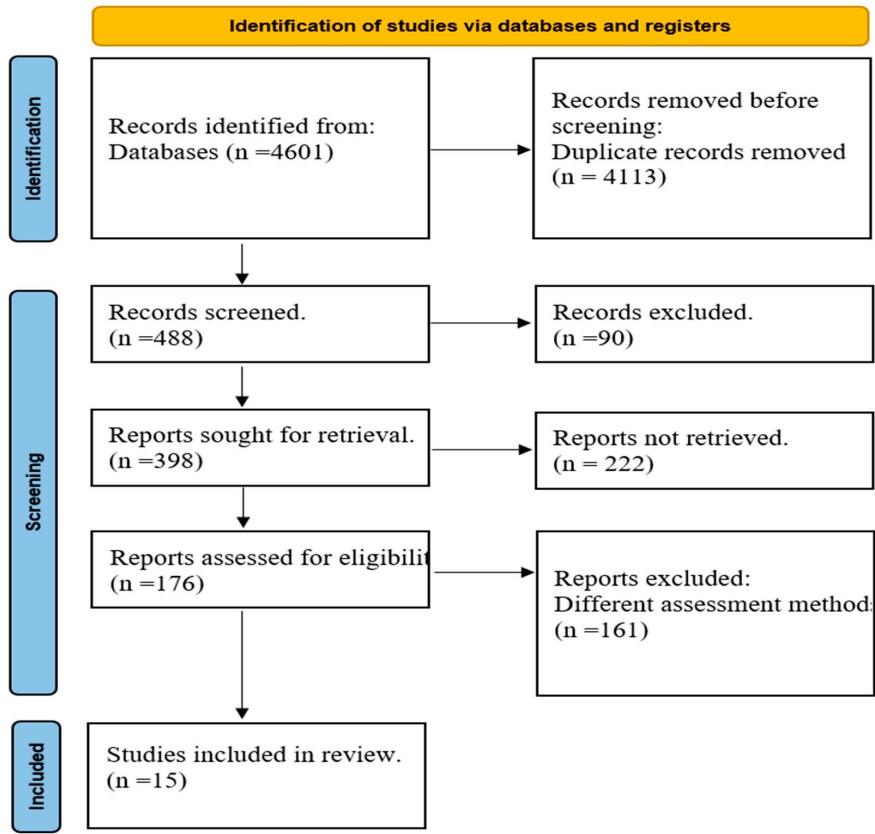

**Figure 1.** PRISMA flow diagram.

*2.4. Assessment of Methodological Quality and Bias*

The methodological quality and risk of bias of all qualifying studies were reviewed separately by three reviewers, namely (M.M.A., O.M.E.R). All studies were evaluated as independent observational cohorts using modified versions of ROBVIS2, a tool developed at the Evidence Synthesis Hackathon. The web application has been developed using the ROBVIS, R package [34]. A consensus was reached to settle discrepancies in the assessment.

*2.5. Data Analysis*

- The data analysis methods of this systematic review include the following methods: Narrative Synthesis: The proposed qualitative technology will include the synthesis and analysis of data from selected studies relating to adoption rates, effectiveness and user perception of digital health treatments in the context of promoting healthy aging. The utilization of narrative synthesis will facilitate a comprehensive comprehension of the implications for both users and healthcare providers.
- The thematic analysis: is used to get and systematically classify similar themes, patterns, and effects observed in a particular study. The present procedure will entail the coding of research outcomes pertaining to the domains of adoption, efficacy, and user experiences in the context of digital health interventions. This study aims to enable a comprehensive examination of the interconnections and diversities within these overarching concepts, thereby offering valuable perspectives on the efficacy and obstacles linked to various digital health interventions in the context of promoting optimal aging.

### 3. Results

*3.1. The Quality Assessment*

This risk of bias assessment provides a thorough evaluation of the internal validity and methodological quality of the 15 studies included in the systematic review [19,35–48]. It

examined each study's risk of bias across six key domains that could influence the reliability and trustworthiness of the results: sequence generation, deviations from intended interventions, missing outcome data, measurement of outcomes, selective outcome reporting, and other potential sources of bias.

Several studies demonstrated minimal risks of bias through strong study designs. Taraldsen et al. (2020) stood out as having the lowest risk across all domains [35]. As a rigorously conducted RCT with adequate randomization procedures, adherence to telehealth protocols, complete follow-up and objective outcome assessment, it represents the highest quality of evidence. Kim et al. (2019), Boot et al. (2016), Sun et al. (2018), Tousignant et al. (2010) and Gatica-Rojas et al. (2023) also employed sound experimental methods with low risks in all areas judged, indicating robust internal validity [36–38,46,47]. Meanwhile, some studies exhibited higher risks of bias in certain domains that undermine confidence in their findings to varying degrees. Zhou et al. (2023) and Perdana et al. (2022) raised major concerns about intervention integrity and subjective outcome measurement biases [41,44]. Frennert et al.'s (2021) issues with sequence generation and missing data introduce ambiguity [42]. Cajita et al. (2018) displayed randomization and reporting biases [48]. Bevilacqua et al. (2021) uniquely faced a clear reporting concern [43]. Ienca et al. (2021) and Chaudhry et al. (2022) also showed multiple "some concerns" ratings [19,39].

This analysis provides a hierarchy of evidence quality that justifies increased certainty in results from studies exhibiting rigorous methodologies and reduced risks of bias versus others warranting more cautious interpretation or further research to address weaknesses compromising validity. It is an invaluable lens for contextualizing this review's findings (Figure 2).

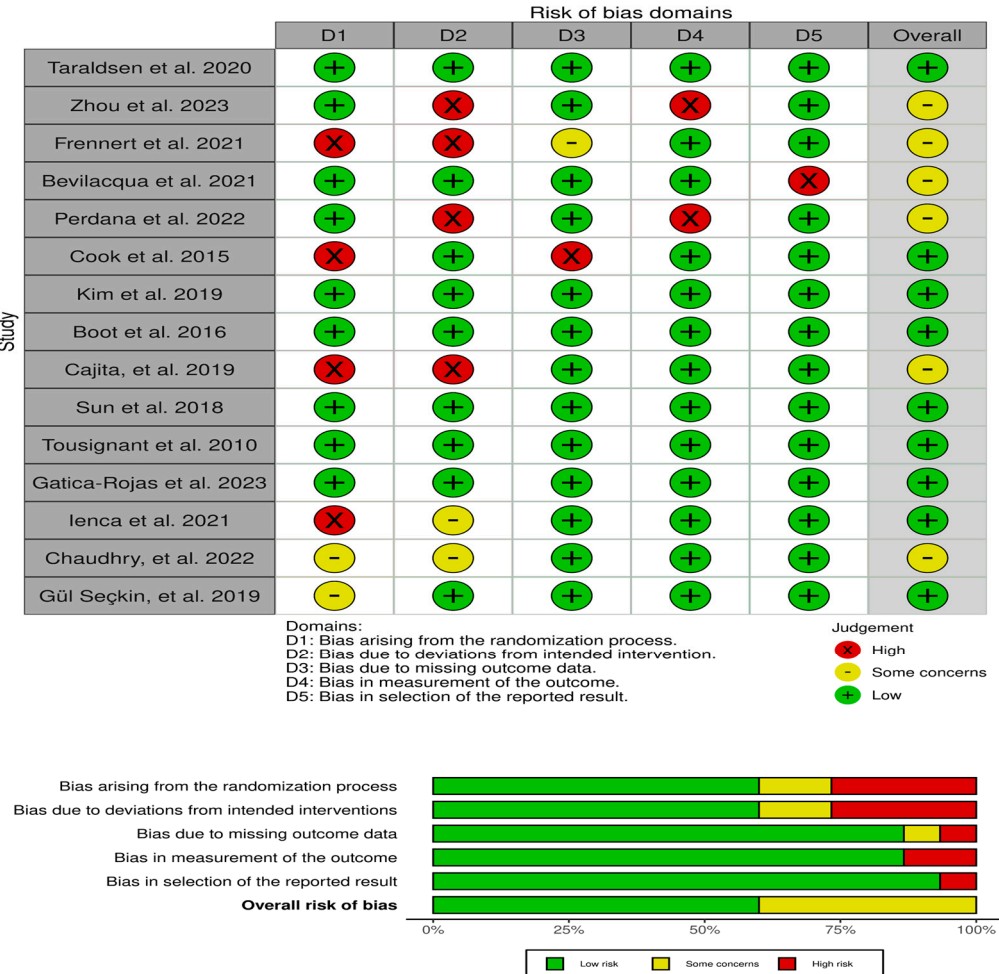

**Figure 2.** Summary of risk of bias [19,35–40,42,44,47,48].

### 3.2. Main Outcomes

Based on the data extracted from the 15 selected articles (Table 1), [19,35–48]. Regarding the study designs included, several studies employ RCTs, which are considered a gold standard in research. They offer controlled conditions to assess interventions' efficacy. Examples include Taraldsen et al.'s assessment of a lifestyle-integrated exercise program and Cook et al.'s web-based health promotion program for older workers [35,45]. Zhou et al.'s study utilizes a cohort design to track trends and variables related to digital health technology use among older adults diagnosed with cancer [41]. This design allows longitudinal assessment and observation. Frennert et al. and Ienca et al. use qualitative approaches to capture perspectives and attitudes. These studies delve into the views, benefits, and drawbacks perceived by older adults regarding welfare technology and digital health interventions, respectively [42].

**Table 1.** The Extraction table.

| Author and Year | Study Design | Title | Participants | Interventions | Results |
|---|---|---|---|---|---|
| Taraldsen et al., 2020 [35] | Randomized Controlled Trial | "Digital Technology to Deliver a Lifestyle-Integrated Exercise Intervention in Young Seniors—The Prevent IT Feasibility Randomized Controlled Trial" | 180 older adults (age 65+) | Evaluate the viability of implementing a Lifestyle-integrated Functional Exercise program and test its efficacy in preventing functional decline in individuals aged 61–70 through the use of digital technology (eLiFE). | Young seniors can safely and effectively participate in a fitness program that incorporates lifestyle changes through the use of information and communication technologies. |
| Zhou et al., 2023 [35] | Cohort Study | "Use of Digital Health Technology Among Older Adults with Cancer in the United States: Findings from a National Longitudinal Cohort Study (2015–2021)" | 1131 older adults (age 60+) | Investigate the patterns and variables linked to the utilization of digital health technologies by cancer patients in their later years. | An increasing number of elderly persons diagnosed with cancer are utilizing digital health technologies, and this trend has been more pronounced during the COVID-19 epidemic. Nonetheless, racial and socioeconomic inequalities persist among cancer survivors in their latter years. Furthermore, there may be some distinct characteristics of digital health technology use among older persons with cancer. |
| Frennert et al., 2021 [42] | Qualitative Analysis | "The concept of welfare technology in Swedish municipal eldercare" | 290 older adults | Talk about how the elderly view welfare technology and what they see as the benefits and drawbacks of using it. | There seems to be an internal struggle among municipal eldercare organizations to bring this goal to fruition, despite the fact that the individuals involved in welfare technology deployment and decision-making are typically rather enthusiastic about it. |
| Bevilacqua et al., 2021 [35] | Quasi experimental design | "eHealth Literacy: From Theory to Clinical Application for Digital Health Improvement. Results from the ACCESS Training Experience" | 58 older | Evaluate the Clinical Application for Digital Health for elderly people | Overall, digital health interventions in geriatric care showed positive outcomes, including reduced hospitalizations and better medication adherence. |

**Table 1.** *Cont.*

| Author and Year | Study Design | Title | Participants | Interventions | Results |
|---|---|---|---|---|---|
| Perdana et al., 2022 [44] | Cross-Sectional Survey | "Seniors' adoption of digital devices and virtual event platforms in Singapore during COVID-19" | 144 older adults (age 60+) | Examine the use of technology by the elderly, employing the social exchange theory as a framework. Additionally, you should look into what drives elderly people to use online event platforms. | The perceived advantages are impacted by social factors and the perceived simplicity of use. When it comes to seniors' plans to use virtual event platforms, these elements are crucial. |
| Cook et al., 2015 [44] | RCT | "A Web-based health promotion program for older workers: randomized controlled trial" | 278 employees aged 50 to 68 | Assess the efficacy of HealthyPast50, an entirely automated web-based health promotion program that targets employees aged 50 and up and is grounded in social cognitive theory. | Older working individuals' short-term food and exercise habits could benefit greatly from an online health promotion program. According to gender impacts, the majority of the positive effects of the program on exercise have been shown in women. |
| Kim et al., 2019 [44] | Cross-sectional study | "Older adults' willingness to share their personal and health information when adopting healthcare technology and services" | 170 elderly | investigate the variables affecting the openness of older persons to sharing their personal and health information through healthcare IT and services, particularly wearable tech and relevant services, with an emphasis on the kind of information requested and the organization making the request. | In order to maintain and improve their health, older persons are careful about the entities asked to share different types of personal and health information when using healthcare technologies and relevant services. |
| Boot et al., 2016 [47] | Randomized Controlled Trial | "The Gamification of Cognitive Training: Older Adults' Perceptions of and Attitudes Toward Digital Game-Based Interventions" | 60 older adults (age 60+) | Investigated the views and feelings of seniors towards game-based interventions following a month of playing digital games (either experimental or control games). | The study's findings emphasize the significance of game design and user experience in encouraging older adults to engage with digital interventions that aim to enhance their cognitive abilities. Specifically, it found that older adults were more motivated to play enjoyable control games rather than gamified brain training interventions, and they had more positive attitudes towards these games overall. |
| Cajita, et al., 2019 [48] | Descriptive, exploratory study | "Facilitators of and Barriers to mHealth Adoption in Older Adults with Heart Failure" | 10 older adults (age 65+) | Investigate how elderly individuals suffering from heart failure see the utilization of mobile devices and discover any variables that can encourage or discourage the uptake of mHealth. | Technology was one of the obstacles that were found. Despite their worries, older folks are open to using mobile health technologies. |
| Sun et al., 2018 [36] | Cross-sectional Survey | "Internet use and need for digital health technology among the elderly: a cross-sectional survey in China" | 669 older adults (age 60+) | Investigates the present situation of senior Internet use, the elements impacting it (including psychological, social, and physical aspects), and the desire for smart services among the aged. | There has to be a greater push to promote digital health technology and lower Internet access restrictions. To ensure that the elderly can enjoy the advantages of internet technology, society, equipment makers, and family members must collaborate. |
| Tousignant et al., 2010 [37] | Randomized Controlled Trial | "A randomized controlled trial of home telerehabilitation for post-knee arthroplasty" | 48 older adults (age 66+) | Comparing the efficacy of conventional rehabilitation after knee replacement surgery with home telerehabilitation | Home telerehabilitation is just as effective as traditional therapy and could open up more treatment options in places with fast Internet. |

**Table 1.** *Cont.*

| Author and Year | Study Design | Title | Participants | Interventions | Results |
|---|---|---|---|---|---|
| Gatica-Rojas et al., 2023 [38] | Randomized Controlled Trial | "Telerehabilitation in Low-Resource Settings to Improve Postural Balance in Older Adults: A Non-Inferiority Randomized Controlled Clinical Trial Protocol" | 60 older adults (age 60+) | Validate the viability and efficacy of a rehabilitation plan targeting the enhancement of postural balance in elderly individuals through the utilization of cost-effective virtual reality. Also, it aims to evaluate two ways of delivering low-cost VR: telerehabilitation (TR) in centers for the elderly and face-to-face (FtF) in rehabilitation facilities. | This study will examine the potential of a low-cost virtual reality (VR) rehabilitation program to enhance postural balance in older adults from a city in Chile that has a sizable rural and underprivileged population. The goal is to provide evidence to inform public health policy decisions. |
| Ienca et al., 2021 [19] | qualitative Study | "Digital health interventions for healthy ageing: a qualitative user evaluation and ethical assessment" | 19 older adults | Investigate the perspectives, requirements, and understandings of older persons residing in the community with reference to the utilization of digital health technology for the promotion of healthy aging. | Digital health technologies were generally well-received by participants, who held the belief that these tools may enhance their overall health, particularly when they were created with the patient in mind. Participants also highlighted safety risks and ethical issues connected to privacy, empowerment, and the absence of human interaction as important factors to consider. |
| Chaudhry et al., 2022 [39] | Experimental Study | "Successful Aging for Community-Dwelling Older Adults: An Experimental Study with a Tablet App" | 25 older adults (age 65+) | explore the viability of an eSenior Care tablet app using the Successful Ageing framework | Older persons from low-income backgrounds regarding the viability and potential influence of a mobile health tool on health-related quality of life. This demographic calls for mHealth support tools and follow-up studies to assess their efficacy. |
| Gül Seçkin, et al., 2019 [40] | Experimental Study | "Digital Pathways to Positive Health Perceptions: Does Age Moderate the Relationship Between Medical Satisfaction and Positive Health Perceptions Among Middle-Aged and Older Internet Users?" | 499 aged 40–93. | Find out how medical satisfaction and good health perceptions are impacted by e-trust, e-health literacy, e-health information seeking, and e-health information consumption. | There were two significant predictors: e-trust and e-health consumption. The positive health perception index was significantly predicted by the e-health literacy and e-trust measures in the OLS regression models. |

Studies by Perdana et al., Kim et al., Sun et al., and Gül Seçkin et al. adopt cross-sectional surveys [36,40,44,46]. They capture a snapshot of the population's views, preferences, and behaviors regarding digital health technology use among older adults. Chaudhry et al., Gatica-Rojas et al., and Boot et al. employ experimental designs. They explore the viability, impact, and perceptions of specific digital health interventions among older adults, offering controlled settings to test interventions' effectiveness.

The cumulative participation of 4671 individuals across these studies reflects a substantial and diverse sample size in exploring digital health interventions among older adults. This sizable cohort contributes to the robustness and breadth of insights into various aspects of technology adoption, efficacy, and user experiences in this demographic.

As regards the main outcomes, the following outcomes were found:

Digital Health Adoption Patterns in the Elderly

The collective findings highlight a nuanced landscape of digital health technology adoption among older adults. Studies by Zhou et al., Perdana et al., Sun et al., and Kim

et al. illuminate the increasing trend of technology adoption among older populations, particularly during critical periods like the COVID-19 pandemic [36,41,44,46]. However, within this growing trend, disparities and inequalities in utilization emerge, as noted by Zhou et al. and Perdana et al., shedding light on variations influenced by racial, socioeconomic, and geographical factors [41,44].

The investigations by Cajita et al., Frennert et al., and Ienca et al. delve into the intricate barriers that impede the widespread adoption of digital health interventions among older adults [42,48]. These barriers encompass technological challenges, concerns about privacy and data security, and organizational hurdles in effectively implementing these solutions within eldercare settings. Insights from these studies emphasize the importance of addressing these barriers to promote wider acceptance and use among older populations.

## Efficacy of Digital Health Interventions for the Elderly and the Health Team

The studies evaluating specific interventions among older adults, such as those conducted by Taraldsen et al., Cook et al., Tousignant et al., Gatica-Rojas et al., and Bevilacqua et al., underscore positive health outcomes [35,37,38,43,45]. These interventions range from lifestyle-integrated exercise programs to web-based health promotion initiatives, demonstrating improvements in physical fitness, rehabilitation, medication adherence, and reduced hospitalizations among older adults.

Furthermore, the emphasis on user-centered design, elucidated by Boot et al. and Ienca et al., highlights the significance of enjoyable, accessible, and user-friendly interventions [19,47]. These studies underscore the importance of considering user experiences and preferences in designing digital health interventions, as these aspects significantly impact older adults' willingness to engage with such technologies.

## Implementing Digital Health in Healthcare Systems

Frennert et al. identified that frontline eldercare staff generally held positive views regarding the potential of technology in supporting their work [42]. However, they also found that various organizational barriers hindered the broader implementation of digital health interventions. This highlights the importance of addressing these barriers to enable successful integration.

One key facilitator identified by the study was managerial support. When managers actively supported and championed the use of technology, it positively influenced the adoption and implementation of digital health interventions. This highlights the significance of engaging leadership and ensuring their commitment to embracing technological advancements in eldercare settings. Optimized workflows that seamlessly integrated digital health tools into routine care were also identified as a facilitator. When technology was incorporated into existing workflows and processes, it enhanced the efficiency and effectiveness of care delivery. This suggests the need for careful consideration of how digital health interventions can be seamlessly integrated into the existing care pathways and routines of healthcare providers.

Additionally, staff training was found to be crucial for successful implementation. Adequate training on using digital health tools and technologies not only improved staff competency but also increased their confidence in utilizing these interventions. It is important to invest in comprehensive training programs that equip healthcare professionals with the necessary skills and knowledge to effectively utilize digital health interventions in their practice.

## Ethical Considerations for Patient-Centric Solutions

Qualitative findings (Ienca et al., Kim et al.) revealed that older adults value privacy, transparency, and preserving human relationships when adopting digital health tools [19,46]. Over-reliance on technology risks isolating seniors or detracting from person-centered care principles. This tension warrants careful balancing of efficiency aims with ethical imperatives of empowerment and trust. Patient-centric collaborative design can help reconcile this balance respectfully.

Ethical considerations related to digital health interventions for older adults are brought to the forefront by Ienca et al. [19]. These considerations encompass privacy concerns, safety risks associated with technology use, and the absence of human interaction in digital interventions. Additionally, longitudinal perspectives provided by studies like Zhou et al.'s longitudinal cohort study offer valuable insights into the evolving patterns of technology adoption among older adults, presenting a comprehensive view of adoption trends over specific periods [41].

## 4. Discussion

This systematic review synthesized evidence on the intersection of digital health interventions and healthy aging, with a focus on adoption patterns, efficacy, and user experience. The 15 studies included provide insights into the feasibility, adoption trends, barriers, facilitators, effectiveness, and qualitative user perspectives surrounding existing digital health solutions for older adults.

Overall, the results indicate that digital health interventions show promise for supporting healthy aging, but thoughtful design and implementation strategies are imperative. Adoption rates, while increasing, remain varied based on user characteristics and intervention factors. Barriers like technological challenges, limited access, and lack of trust must be addressed. Efficacy data, though still emerging, suggest benefits for outcomes like reduced hospitalizations, better chronic disease management, and improved balance. However, enjoyment, ease of use, and patient-centricity are key for user acceptance and engagement.

### 4.1. Adoption Rates of Digital Health Interventions

One key aspect of digital health interventions is their adoption among older adults. While older adults may face barriers such as limited technological literacy and access to digital devices, studies have shown a gradual increase in their acceptance and use of digital health technologies [49]. For instance, a systematic review by Moffatt et al. (2023) found that older adults are willing to adopt digital health interventions, particularly when they perceive them as useful, easy to use, and beneficial for their health [50]. Moreover, the COVID-19 pandemic has accelerated the adoption of telehealth and remote monitoring technologies among older adults, as these interventions offer safe and convenient alternatives to in-person healthcare visits [51].

However, it is important to acknowledge that there is still a digital divide among older adults, with disparities in access to and proficiency in using digital technologies. Certain subgroups, such as those with lower socioeconomic status or living in rural areas, may face greater challenges in adopting digital health interventions [52]. Bridging this divide requires targeted efforts to improve digital literacy, provide training programs, and ensure equitable access to technology and internet connectivity [53].

### 4.2. Efficacy of Digital Health Interventions

The effectiveness of digital health interventions in promoting healthy aging has been investigated across various domains, including chronic disease management, preventive care, and mental health support [54]. Overall, the evidence suggests that digital health interventions can have positive impacts on health outcomes among older adults [55].

Studies have shown that telehealth interventions can improve chronic disease management and self-care behaviors in older adults. Remote monitoring devices and mobile apps enable continuous monitoring of vital signs, medication adherence, and lifestyle factors, allowing healthcare providers to intervene in real time and provide personalized feedback [56–58]. This proactive approach has been found to enhance disease control, reduce hospital admissions, and improve the quality of life in older adults with conditions such as diabetes, hypertension, and heart failure [59].

Digital health interventions also play a crucial role in preventive care for older adults. Mobile apps and wearable devices can facilitate physical activity tracking, fall prevention, and medication reminders, promoting healthy behaviors and reducing the risk of adverse

health events [60]. Furthermore, digital interventions that provide health education, screening tools, and decision support can empower older adults to make informed choices about preventive measures, such as vaccinations and cancer screenings [61,62].

In the realm of mental health, digital interventions have shown promise in addressing the psychological well-being of older adults. Mobile apps and online platforms offer mental health resources, cognitive training programs, and virtual support groups, providing accessible and convenient avenues for older adults to manage stress, anxiety, and depression [63]. Additionally, telepsychiatry and teletherapy can overcome barriers to mental healthcare access, particularly for individuals residing in underserved areas or facing mobility limitations [64].

### 4.3. User Experience of Digital Health Interventions

User experience encompasses various dimensions, including usability, acceptability, and satisfaction, and plays a crucial role in the successful implementation and long-term engagement with digital health interventions among older adults [65]. Several factors influence user experience, including design features, ease of use, perceived usefulness, and personalization. Usability is a key consideration in the design of digital health interventions for older adults [66]. User interfaces should be intuitive, with clear navigation, legible text, and appropriate font sizes [67]. Incorporating user-centered design principles and conducting usability testing with older adult populations can ensure that digital interventions are accessible and easy to navigate [68].

Acceptability and satisfaction are influenced by factors such as perceived usefulness, perceived ease of use, and perceived benefits. Older adults are more likely to adopt and engage with digital health interventions when they perceive them as valuable additions to their healthcare routines and as tools that enhance their overall well-being [69]. Personalization of interventions based on individual needs, preferences, and health goals can also contribute to higher acceptability and satisfaction among older adults [70].

### 4.4. Digital Health Adoption Patterns

This review reveals that while the adoption of digital health technologies has gradually increased among older adults, uptake remains varied across demographic groups and intervention types. Facilitators like perceived usefulness, social influences, and ease of use can promote adoption [50,71,72]. However, barriers spanning technological challenges, limited accessibility, lack of awareness, and concerns over privacy and trust persist, especially among disadvantaged groups [73–75].

During the COVID-19 pandemic, the adoption of telehealth and remote monitoring rose out of necessity, indicating that external circumstances can accelerate uptake [76]. However, many studies in this review preceded the pandemic. Future research should examine whether these crisis-driven gains are sustained long-term. Meanwhile, usage divides and reluctance persist for emerging technologies like wearables, highlighting the need for research on adoption facilitators specific to these tools [77].

Understanding older adults' perspectives is key. They exhibit selective sharing of health information depending on entity and purpose, valuing privacy and transparency [78]. Many have positive attitudes if digital health solutions are designed in a patient-centric manner, reflecting their needs and capacities [58]. The involvement of end-users, caregivers and clinicians in co-design can ensure that tools align with preferences and lifestyles [58].

### 4.5. Efficacy of Digital Health Interventions

This review indicates that digital health interventions can improve outcomes like medication adherence, hospitalizations, dietary and exercise practices, and balance [60,61]. However, efficacy data remain limited. Rigorous randomized controlled trials with clinically meaningful endpoints are needed to provide robust evidence and elucidate the mechanisms of benefit [55]. Future research should assess impacts on quality of life, clinical outcomes, healthcare utilization, and costs.

Findings suggest that web-based health promotion and disease management programs have the potential to enhance prevention and self-care practices if designed appropriately for the target population [63,64]. Telerehabilitation and telehealth models also appear effective for rehabilitation and chronic disease management, expanding access to care [65]. Virtual reality-based tools show promise for improving balance and mobility [66].

However, success hinges on thoughtful design. Interactive programs with reminders and peer support may increase engagement [67]. Entertainment-based programs focusing on enjoyment over health promotion can avoid stigma and increase acceptance [68]. Co-designing interventions with end users and iteratively testing acceptability can help ensure tools align with older adults' needs and interests prior to definitive trials [69].

### 4.6. Implementing Digital Health in Healthcare Systems

Findings reveal that while frontline staff may perceive the benefits of digital health, organizational barriers can hinder implementation [70]. Managerial support, training, and workflows integrating technologies into routine care are key facilitators [71]. Global policy frameworks also emphasize the need for coordinated deployment and capacity building for eHealth [72].

As tools transition from efficacy studies to real-world implementation, additional research on organizational factors influencing adoption and sustainability is critical. Hybrid effectiveness-implementation trials can evaluate clinical impacts while gathering data on implementation barriers, facilitators and strategies [73]. Evidence on optimal service models integrating technologies like telehealth into chronic disease management programs will guide scale-up. Health economic assessments can strengthen the business case [74].

### 4.7. Ethical Considerations for Patient-Centric Solutions

Older adults value digital health technologies designed with transparency, security, and patient partnership in mind [75]. But tensions exist between improving care and respecting privacy. Striking an ethical balance is critical so that gains do not come at the expense of empowerment or autonomy [76].

Regulations like the EU's General Data Protection Regulations provide data protection frameworks, but additional protections may be needed for vulnerable groups. Transparent consent processes and data governance policies can help earn patient trust [77]. User control over health data sharing preferences allows customization, balancing desired benefits.

There are also concerns that over-reliance on technology could isolate older adults or detract from human relationships in care. Solutions that thoughtfully combine digital tools with interpersonal support may mitigate such risks. Centering ethics, patient values and shared decision-making in design can help ensure sustainable adoption [78].

### 4.8. Methodological Considerations and Limitations

The manuscript presents a comprehensive evaluation of digital health interventions for promoting healthy aging among older adults. Through a systematic search of major databases, the authors ensure a robust coverage of relevant studies. The inclusion of diverse study designs allows for a comprehensive analysis of adoption patterns, efficacy, and user experience. One limitation of the study is the potential for publication bias. The authors relied on published literature from selected databases, which may introduce a bias toward studies with positive or significant findings. This could lead to an overrepresentation of studies that demonstrate the effectiveness or positive impact of digital health interventions for promoting healthy aging while potentially excluding studies with null or negative results.

The manuscript acknowledges the preliminary nature of efficacy data and emphasizes the need for more rigorous trials. It also highlights barriers and divides related to technology challenges and distrust among older adults, emphasizing the importance of improving accessibility and acceptance. While most studies focused on high-income settings, the manuscript recognizes the need for broader inclusion of populations and settings. The

manuscript incorporates qualitative studies to enhance the understanding of digital health interventions for promoting healthy aging. Qualitative studies provide valuable insights into the subjective experiences, perceptions, and preferences of older adults regarding these interventions. By including qualitative data, the manuscript captures the lived experiences and perspectives of users, shedding light on factors influencing adoption, user satisfaction, and potential barriers. This qualitative component adds depth and richness to the overall analysis, complementing the quantitative evidence and providing a more comprehensive understanding of the user experience and its impact on the effectiveness of digital health interventions. Overall, the manuscript demonstrates a methodologically sound approach, acknowledging limitations and contributing to the evidence-based dialogue on digital health interventions for healthy aging.

*4.9. Future Research and Recommendations:*

Based on the results of the systematic review, the following recommendations were set:

- Conduct more rigorous randomized controlled trials to establish efficacy evidence for different types of digital health interventions in promoting healthy aging. Outcomes should include clinically meaningful measures of health, functioning, quality of life, healthcare utilization and costs;
- Longitudinal research is needed to evaluate long-term impacts, effectiveness over time as technologies and needs change, and sustainability of adoption into routine care and lifestyles;
- Comparative effectiveness research can elucidate which digital solutions work best for specific conditions, populations and healthcare systems/models.
- Studies should include broader populations from diverse settings and backgrounds to ensure equitable representation beyond current high-income-focused research;
- Further qualitative research can deepen understanding of user experiences, needs, perspectives on ethics/trust and optimization of technologies based directly on end-user feedback;
- Implementation science studies are warranted to evaluate real-world integration into care delivery, identify barriers/facilitators, and test strategies for scalable adoption addressing organizational factors;
- Health economic analyses can strengthen the case for investment and reimbursement by quantifying cost offsets from reduced hospitalizations, complications and informal caregiving needs;
- Policy research should explore approaches for addressing digital and health literacy divides, accessibility standards, data governance frameworks, and workforce training/skills.

## 5. Conclusions

This systematic review highlights the potential of digital health interventions in promoting healthy aging among older adults. The findings demonstrate that these interventions, including wearable devices, telehealth platforms, mobile health apps, and personalized medicine enabled by big data analytics and AI, can enhance access, monitoring, and self-care for older populations.

The study reveals that the adoption rates of digital health technologies have increased, particularly during the COVID-19 pandemic. However, certain barriers and divides, such as technology challenges and distrust, still need to be addressed to ensure equitable access and utilization of these interventions.

Web-based programs and telerehabilitation have shown promising benefits in promoting healthy behaviors and balance among older adults. Users generally have positive attitudes towards digital health interventions but emphasize the importance of patient-centric and ethical design. It is crucial to consider the individual needs and capacities of older adults in the implementation of these technologies to maximize their effectiveness.

While the efficacy data for digital health interventions in promoting healthy aging are still preliminary, there is a clear need for more rigorous trials to establish their effectiveness. Additionally, research focusing on real-world integration, ethics, and user experiences of digital health technologies targeting older populations is warranted.

To fully harness the potential of digital health interventions for promoting healthy aging, thoughtful implementation strategies and evidence-based, patient-centered solutions are crucial. These interventions have the potential to facilitate aging in place, reduce healthcare resource utilization, and enable early detection of health status changes. However, it is essential to address the challenges faced by older adults, including age-related barriers, disparities in access, and individual differences in preferences and needs.

This systematic review underscores the importance of digital health interventions in supporting healthy aging. While further research is needed, the existing evidence suggests that digital health technologies can play a significant role in improving the health, well-being, and quality of life of older adults. By considering user needs, addressing barriers, and ensuring equitable access, digital health interventions can be optimally designed to maximize adoption, efficacy, and positive user experiences.

**Supplementary Materials:** The following supporting information can be downloaded at: https://www.mdpi.com/article/10.3390/su152316503/s1. PRISMA 2020 Checklist [79].

**Author Contributions:** Conceptualization, M.M.A. and M.S.; methodology, M.S.; software, M.S.; validation, M.M.A., M.S. and O.M.E.R.; formal analysis, M.S.; investigation, M.M.A.; resources, M.M.A.; data curation, M.M.A.; writing—original draft preparation, M.M.A.; writing—review and editing, M.M.A.; visualization, M.M.A.; supervision, O.M.E.R.; project administration, O.M.E.R.; funding acquisition, M.S. All authors have read and agreed to the published version of the manuscript.

**Funding:** This research received no external funding.

**Institutional Review Board Statement:** Not applicable.

**Informed Consent Statement:** Not applicable.

**Data Availability Statement:** Data are available upon request.

**Conflicts of Interest:** The authors declare no conflict of interest.

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
