# Peer review of "Digital Health Interventions for Promoting Healthy Aging: A Systematic Review of Adoption Patterns, Efficacy, and User Experience"

_sustainability, doi:10.3390/su152316503_

Round 1
Reviewer 1 Report
Comments and Suggestions for Authors
The study's research question lacks specificity and clarity. It does not provide a clear framework for evaluating the intersection of digital health interventions and healthy aging. A more precise and focused research question is needed. The introduction section is vague and fails to establish the significance of the study. It lacks a clear rationale for why the research question is important and what gap in the literature it addresses. The methodology is not well-defined. The inclusion criteria for studies are broad, and it is unclear how the selected studies were relevant to the research question. The lack of a comparison group (as indicated in the PICO framework) raises concerns about the validity of the results. The literature review is insufficient, and the study fails to provide an adequate background on the topic. There is no discussion of the existing knowledge on digital health interventions for healthy aging or their impact. The presentation of results is incomplete. While the study mentions that 15 studies were included, it provides minimal information about the findings of these studies. More detailed information on the outcomes, findings, and their implications is necessary. The conclusions do not align with the findings presented in the study. The manuscript suggests that digital health interventions have "significant potential," but the study's methodology and results do not support such a claim. The conclusions should be more grounded in the evidence presented. The study lacks a critical analysis of the quality of the included studies. It is essential to assess the methodological rigor of the studies to determine the reliability of the findings. The manuscript would benefit from a more structured approach, including a clear problem statement, specific objectives, and a well-defined research methodology. The current organization lacks coherence and clarity. The manuscript does not discuss the limitations of the study adequately. It is essential to acknowledge the limitations, especially in terms of the search strategy, inclusion criteria, and potential biases. The references need to be properly formatted and should include a more comprehensive selection of relevant literature to support the study's claims. The manuscript lacks a clear plan for future research or recommendations for improving the field of digital health interventions for healthy aging. Providing suggestions for further investigation or practical implications is essential. The writing quality and organization of the manuscript need improvement. The language is often unclear, and the content is disorganized.
Comments on the Quality of English LanguageMinor Changes Required
Author Response
Dear Reviewer,
Thank you for your valuable feedback on our manuscript titled "Digital Health Interventions for Promoting Healthy Aging: A Systematic Review of Adoption Patterns, Efficacy, and User Experience." We appreciate your thorough evaluation of the study and have carefully considered your comments. We have made significant revisions to address the concerns raised, and we believe these changes have strengthened the clarity, focus, and overall quality of the manuscript. Below, we provide a point-by-point response to each of your comments:
Research Question: We agree that the research question needed more specificity and clarity. We have revised the research question to clearly evaluate the intersection of digital health interventions and healthy aging, focusing on adoption patterns, efficacy, and user experience among older adults. The revised research question is presented in the introduction section.
Significance and Rationale: We have enhanced the significance of the study by providing a clearer rationale for why the research question is important and the gap it addresses in the literature. We have included additional information on the challenges posed by global population aging and the potential benefits of digital health interventions for older adults. The introduction section now better establishes the context and significance of the study.
Methodology: We acknowledge that the methodology section required more clarity. We have provided a more detailed description of our search strategy and selection criteria, addressing the issue of broad inclusion criteria. We have also clarified the relevance of the selected studies to the research question. While we primarily focused on observational and qualitative studies, we have included a discussion on the limitations of not having a comparison group. This provides transparency and acknowledges the potential impact on the validity of the results.
Literature Review: We have revised the literature review section to provide a more comprehensive background on digital health interventions for healthy aging. We discuss the existing knowledge in the field and highlight the impact these interventions can have on various aspects of older adults' health. This revision strengthens the contextual understanding of the topic.
Presentation of Results: We apologize for the incomplete presentation of results. We have provided more detailed information on the outcomes, findings, and implications of the included studies. The revised manuscript now offers a more comprehensive overview of the results obtained from the selected studies.
Conclusions: We acknowledge the discrepancy between the conclusions and the presented findings. The revised conclusions have been grounded in the evidence presented in the study. We have rephrased the conclusions to align with the methodology and results, emphasizing the potential of digital health interventions while also acknowledging the need for further research and patient-centered design optimizations.
Critical Analysis of Study Quality: We appreciate your suggestion regarding the assessment of the methodological rigor of the included studies. We have conducted a critical analysis of the study quality and have included a discussion on this aspect in the methodology section. This analysis provides insights into the reliability of the findings and strengthens the study's validity.
Structure and Organization: We have restructured the manuscript to improve coherence and clarity. We have included a clear problem statement, specific objectives, and a well-defined research methodology. The revised manuscript follows a logical flow, ensuring that the content is well-organized and easy to follow.
Limitations: We have expanded the discussion on the limitations of the study. We now explicitly acknowledge the limitations related to the search strategy, inclusion criteria, and potential biases. This revision enhances the transparency of the study and provides a more comprehensive understanding of its limitations.
References: We have thoroughly reviewed and properly formatted the references. We have also included a more comprehensive selection of relevant literature to support the claims made in the study. The revised manuscript now provides a robust foundation of supporting evidence.
Future Research and Recommendations: We have addressed the need for suggestions for further research and practical implications. The revised manuscript includes a section on future research directions and recommendations for improving the field of digital health interventions for healthy aging. This addition strengthens the manuscript by highlighting avenues for future investigation and practical application.
Writing Quality and Organization: We have extensively revised the manuscript to improve the writing quality and organization. We have clarified the language, improved sentence structure, and enhanced the overall readability. The content is now presented in a more organized manner, ensuring coherence throughout the manuscript.
Once again, we sincerely appreciate your valuable feedback, which has significantly contributed to the improvement of our manuscript. We are confident that the revised version addresses the concerns raised and enhances the overall quality of the study. We hope that you find the changes satisfactory and consider the manuscript suitable for publication.
Thank you for your time and consideration.
Reviewer 2 Report
Comments and Suggestions for Authors
This systematic review evaluates the intersection of digital health interventions and healthy aging. In the context of global aging and the development of digital technologies, this review is important. Before publication, I have two suggestions:
1. Did the authors get enough articles? In the end, only 15 studies were included. In the studies, there is one study on e-health literacy, but there are maybe twenty or thirty studies on e-health literacy in older people. Therefore, authors should emphasize the strategy of article selection.
2. The review doesn't show any data. If possible, in the results and discussion, there should be some data related to the main findings, so that readers can easily understand.
Author Response
Response to Reviewer Comments:
We would like to thank the reviewer for their valuable feedback on our systematic review. We have carefully considered the suggestions and would like to address them below:
Article Selection Strategy: We acknowledge the concern raised about the number of included studies in our review. While it is true that the final number of included studies was limited (15 studies), we would like to highlight that our aim was to provide a comprehensive analysis of the available evidence within the scope of our research question. We conducted a systematic search of three major databases and followed a rigorous screening process to select studies that met our inclusion criteria. However, we acknowledge that the selection of articles can be influenced by various factors, including the search strategy and database coverage. We have revised the manuscript to emphasize the article selection strategy and have provided a clear rationale for the final inclusion of studies.
Presentation of Data: We appreciate the suggestion to include more data in the results and discussion section to enhance readers' understanding. While the focus of our review was on synthesizing and analyzing the findings from the selected studies, we understand the importance of presenting data to support the main findings. In response to this suggestion, we have revised the manuscript to include relevant data, such as study characteristics, intervention outcomes, and user feedback, where applicable. This addition will provide readers with a clearer overview of the main findings and facilitate a better understanding of the results.
Once again, we appreciate the reviewer's comments and suggestions. We believe that the revisions we have made to the manuscript address these concerns and enhance the clarity and comprehensibility of our review.
Reviewer 3 Report
Comments and Suggestions for Authors
That´s a good article.
The redaction of article is very good.
The quality of the writing is good and the content presents a hight quality
There are quality in figures and table, the reference list cover the literatura adequately and the methods are sufficiently.
In general, the literatura review is perfect, the results and conclusions are good.
Comments on the Quality of English LanguageIts a good english
Author Response
Thank you very much for your positive feedback on the quality of the article. We greatly appreciate your kind words regarding the writing, content, figures, tables, reference list, and methods. We are pleased to hear that the literature review, results, and conclusions are deemed satisfactory.
We have devoted considerable effort to ensure that the manuscript is comprehensive, well-written, and methodologically sound. Your positive assessment of these aspects is encouraging and validates our work.
We sincerely appreciate your time and valuable feedback. Your positive evaluation motivates us to continue producing high-quality research in the field. Should you have any further comments or suggestions, please do not hesitate to let us know.
Reviewer 4 Report
Comments and Suggestions for Authors
The manuscript entitled “Digital Health Interventions for Promoting Healthy Aging: A Systematic Review of Adoption Patterns, Efficacy, and User Experience” is the study aiming to summarize the literature on digital health interventions for older adults (50+) on various health topics, evaluating adoption metrics, clinical effectiveness, and qualitative user experiences
The work has an interesting topic, it is important in the field of healthy aging and especially for the generations that will soon be in their 50s and are much more connected to gadgets and their possibilities.
The manuscript was written according to the guidelines and instructions for the systematic review (in some parts even too detailed).
What is not very clear is in line 189 and Figure 1, can you clarify how the assessment methods differed so that you only got 15 suitable papers.
Standardize the citation of references in the manuscript, e.g. line 217 - numbers in square brackets should be used.
Perhaps it would be good if some main aims were linked so that each selected paper doesn’t represent an individual aim.
Perhaps the concept of the whole manuscript should follow the discussion section - Digital Health Adoption Patterns; Efficacy of Interventions; Implementing in Healthcare Systems.
The shortcoming of the manuscript is the small number of papers and in parallel the large number of listed "objectives" of technology use, and in the future it should be a bit better/clearer to systematize the data in terms of a final conclusion.
Comments on the Quality of English Language
English language is fine. Minor editing is required.
Author Response
We would like to express our gratitude to the reviewer for their valuable feedback on our manuscript titled "Digital Health Interventions for Promoting Healthy Aging: A Systematic Review of Adoption Patterns, Efficacy, and User Experience." We appreciate your positive comments on the relevance of the topic and the importance of digital health interventions for healthy aging.
Regarding your specific comments, we would like to address them as follows:
Clarification of Assessment Methods: We apologize for any confusion caused by the lack of clarity in line 189 and Figure 1 regarding the assessment methods. In our systematic review, we employed a comprehensive search strategy to identify relevant studies. The assessment methods differed across the included studies due to the diversity of interventions and outcomes examined. We have revised the manuscript to provide more explicit clarification of the assessment methods and how they varied among the selected studies.
Standardization of References: Thank you for pointing out the inconsistency in the citation of references. We acknowledge the importance of using standardized citation formats. We have carefully reviewed the manuscript and made the necessary revisions to ensure that the references are consistently cited using numbers in square brackets throughout the text.
Linking Main Aims and Systematizing Data: We appreciate your suggestion to link the main aims of the selected papers to provide a clearer structure and avoid representing each paper as an individual aim. We have revised the manuscript to improve the organization and coherence of the aims and findings. By linking the main aims, we aim to provide a more cohesive narrative that highlights overarching themes and conclusions derived from the collective evidence.
Conceptual Structure of the Manuscript: We acknowledge your observation regarding the conceptual structure of the manuscript. We agree that organizing the manuscript to follow the discussion sections on Digital Health Adoption Patterns, Efficacy of Interventions, and Implementing in Healthcare Systems could enhance the clarity and flow of the manuscript. We have revised the manuscript accordingly to align with this suggested structure.
Systematizing Data and Final Conclusion: We appreciate your feedback regarding the systematization of data and the need for a clearer final conclusion. We have taken your suggestion into account and have worked on better organizing the presented data and drawing more concise and informative conclusions. The revised manuscript now provides a more systematic and comprehensive overview of the findings, resulting in a clearer final conclusion.
Once again, we would like to thank the reviewer for their valuable comments and suggestions. We believe that the revisions we have made address these concerns and improve the clarity and structure of the manuscript. We greatly appreciate your time and feedback.
Round 2
Reviewer 1 Report
Comments and Suggestions for Authors
Accept